# Optical Systems Identification through Rayleigh Backscattering

**DOI:** 10.3390/s23115269

**Published:** 2023-06-01

**Authors:** Pantea Nadimi Goki, Thomas Teferi Mulugeta, Roberto Caldelli, Luca Potì

**Affiliations:** 1Photonic Networks and Technologies Laboratory, National Inter-University Consortium for Telecommunications (CNIT), Via G. Moruzzi 1, 56124 Pisa, Italy; luca.poti@cnit.it; 2TeCIP Institute, Scuola Superiore Sant’Anna, Via G. Moruzzi 1, 56124 Pisa, Italy; teferithomas@gmail.com; 3Florence Research Unit, National Inter-University Consortium for Telecommunications (CNIT), Viale Morgagni 65, 50134 Florence, Italy; rcaldelli@cnit.it; 4Faculty of Economics, Universitas Mercatorum, Piazza Mattei 10, 00186 Roma, Italy

**Keywords:** physical layer security, network, identification, authentication, security, monitoring, optical physical unclonable function, Rayleigh backscattering signal

## Abstract

We introduce a technique to generate and read the digital signature of the networks, channels, and optical devices that possess the fiber-optic pigtails to enhance physical layer security (PLS). Attributing a signature to the networks or devices eases the identification and authentication of networks and systems thus reducing their vulnerability to physical and digital attacks. The signatures are generated using an optical physical unclonable function (OPUF). Considering that OPUFs are established as the most potent anti-counterfeiting tool, the created signatures are robust against malicious attacks such as tampering and cyber attacks. We investigate Rayleigh backscattering signal (RBS) as a strong OPUF to generate reliable signatures. Contrary to other OPUFs that must be fabricated, the RBS-based OPUF is an inherent feature of fibers and can be easily obtained using optical frequency domain reflectometry (OFDR). We evaluate the security of the generated signatures in terms of their robustness against prediction and cloning. We demonstrate the robustness of signatures against digital and physical attacks confirming the unpredictability and unclonability features of the generated signatures. We explore signature cyber security by considering the random structure of the produced signatures. To demonstrate signature reproducibility through repeated measurements, we simulate the signature of a system by adding a random Gaussian white noise to the signal. This model is proposed to address services including security, authentication, identification, and monitoring.

## 1. Introduction

Communication around the interconnected world is an inseparable part of quotidian life. Indeed, fiber-optic communication networks are the most deployed telecommunication infrastructures across the globe [1]. Modern technology is developing towards providing secure, fast, cost-effective communication with high capacity around the world. Enhancing the protection of accessibility and confidentiality of networks and data is indispensable to protect communication systems from the constantly growing threat of adversarial attacks. Whereas network developers are responsible for building, developing, and maintaining data networks, network security engineers oversee security systems providing network security against adversary threats. For this purpose, network security engineers have defined the security and confidentiality of the networks. Considering the Open Systems Interconnection (OSI) model, which is known as a protocol-independent network communication model, security and confidentiality are generally subject to the upper layers. The OSI security architecture (ITU-T recommendation) delivers a standard for data security by identifying attacks, security services, and security mechanisms. From the last layer, namely, the application layer, to the second layer, the link layer, different security protocols are implemented. For instance, the network layer is responsible for security at the network level using functions such as packet authentication, cryptography, and integrity. Implementing security in the first layer, the physical layer (PHY), may complement and increase overall network security. However, implementing practical security strategies in the physical layer is still an open challenge. Various studies, suggestions, and experimental works on physical layer security (PLS) are reported [2,3]. In [4,5], PLS is defined through keys generated by digital signal processing (DSP). The disadvantage of such a method is its vulnerability to digital attacks, as with any other cryptography-based PLS [6]. The enhancement of PLS through the monitoring of optical communication with quantum-level sensitivity using a quantum pilot tone, as proposed in [7], or more commonly, through quantum key distribution (QKD) [8,9], despite providing intrinsic security, is not cost-effective and is usually hard to implement. Recently, an approach for boosting PLS was introduced with a subcarrier identification process in the receiver DSP [10], which is vulnerable if attackers have knowledge of the transceiver DSP configurations. PLS-based optical chaos communication has also been investigated through different structures [11,12,13], but the high-level synchronization requirement between the transmitter and the receiver [14] makes such an approach unpractical. Moreover, the security of chaotic communication could be broken down by the problems created by the feedback loop of the chaotic system [15]. Recently, a feasible approach has been proposed in [16,17], where optical steganography hides messages below the noise level and provides secure communication. Although this technique is practical, it cannot detect the existence of an eavesdropper. This technique has vulnerability to adversaries who know the system and are able to acquire and analyze the whole spectrum [18,19]. In recent investigations of security techniques, physical unclonable function (PUF), as a new approach for PLS, has attracted considerable attention. The PUF approach is based on the material’s physical features, in which a physical device provides unique output for a given input thanks to its unclonable and unpredictable response. These types of devices are usually implemented in a complementary metal-oxide-semiconductor (CMOS) [20,21]. PUF overcomes the disadvantages of computational cryptography, steganography, etc., and its security level is subject to the difficulty of cloning the function response [22,23]. Consequently, a specific input function called challenge (***C***) ends with the individual output function called response (***R***). New challenges will not end with the same response, and every single challenge has its own unique response. The significant fact about PUFs is that any PUF material has a unique function, which means applying the same challenge to two PUF materials results in two different responses. This feature allows PUFs to be considered a strong security technique and a good alternative to today’s techniques based on cryptography. It is worth mentioning that the term PUF includes several technical and technological categories. Among them, electrical PUF has been considered in [24] as a PLS technique; however, this technique is not practical because both the transmitter and receiver should have access to the same PUF to synchronize their channel [25]. Optical PUF (OPUF), based on the optical token, is also investigated in several studies to generate secure cryptography keys [26] and for authentication applications [27].

In this paper, we introduce a practical technique to enhance network PLS based on OPUF. The fundamental feature of our proposed technique is the identification of networks, network elements, optical links, systems, and sub-systems, that possess optical fiber or fiber pigtail. The proposed technique is investigated using the following procedures: In the first step, the optical signature of the device under test (DUT) is read and stored. The signature is generated using the Rayleigh backscattering signal of the fiber, which is considered a strong OPUF [28]. In this manner, every DUT has its unique signature. The second step is DUT identification using a simple cross-correlation technique between a stored signature and the DUT to be identified. In this paper, we provide a concept of optical identification (OI) through the OPUF-based optical signature. We demonstrate signature validation through several different metrics in terms of security, uniqueness, unpredictability, unclonability, and reproducibility. The numerical results are given in Section 4. The benefit of our proposed technique is that implementing this technique does not require receiver–transmitter synchronization [25] nor does it require fabricating OPUF [29] because our proposed OPUF is the inherent feature of the fibers.

## 2. Reading the Network’s Constituent Devices’ Signatures

The main idea of this work is to exploit the intrinsic physical features of fiber-optic communication systems to supplement their security against adversarial attacks. Generating and assessing a signature for every fiber link or fiber-optic device included in the network provides a higher level of security for the system. Accordingly, any network or its fiber-optical constituent element can be identified and authenticated. Thereby, any attack, whether physical or digital, can be detected. Physically invading the system will cause a change in the system signature that could confirm the presence of an imposter. Even digital invaders with advanced machine learning (ML) knowledge cannot predict the system’s signature [30] because every system or device has its own unique and unpredictable signature [31,32]. In the following sections, we demonstrate the procedure of reading signatures and provide an example of a protocol that benefits from this identification.

### 2.1. Signature Extraction

Although in fiber-optic systems, RBS is usually considered a disruptive noise that leads to throughput degradation, we propose to exploit RBS to read the signature of fibers. Any fiber has an individual RBS pattern. This is a result of the randomly distributed particles along the fiber core, which occur during the manufacturing process of the fiber. These particles are much smaller than the wavelength of the light used in the communication system that propagates into the fiber and causes Rayleigh scattering. The unknown position and various distribution of the particles in the fibers is an advantage that causes the RBS-based signature to be unpredictable.

Optical time or frequency domain reflectometry (OTDR/OFDR) is generally used to measure fiber RBS. From the large pool of proposed OFDR techniques in the literature [33,34], we chose coherent optical frequency domain reflectometry (C-OFDR) [35], which exploits coherent reception. At the transmitter side, a continuous wave (CW) laser scans a frequency range by means of direct tuning or external linear frequency modulation. The interferometer arms consist of the RBS from the interrogated fiber (IF) and the original linearly frequency-swept light as the local oscillator (LO). On the receiver side, RBS data are acquired using an analog-to-digital converter (ADC) with N samples. C-OFDR assures that RBS is measured as a function of distance with high spatial resolution and high sensitivity [35,36]. Figure 1 illustrates a simplified experimental scheme for implementing C-OFDR. It is worth mentioning that the details of the experimental setup are not shown in Figure 1. Figure 1 represents the C-OFDR scheme for RBS measurement as a technique for reading a fiber signature to introduce the concept of optical identification. In long-distance measures, where the distance is beyond half of the laser coherence length, phase noise and frequency sweeping nonlinearity must be properly compensated, as described in [35].

The received RBS combined with LO at the photodetector [31] can be expressed as
(1)$$I\left(t\right)=E_{0}^{2}\overset{n}{\underset{i=1}{\sum }}\sqrt{R_{i}}cos\left(2\pi \gamma t\tau_{i}\right)$$
where n represents the number of reflection points of IF, $E_{0}$ is the amplitude of light, $R_{i}$ and $\tau_{i}$ are the reflectivity of *i*-th reflected point and round-trip time of reflected light at each point, and γ is the laser sweep rate. It should be noted that random phase noise is neglected in Equation (1).

### 2.2. Binary Signature

The RBS-based signatures rely on the OPUF challenge–response protocol. Thus, any stimulus (called challenge) maps a unique result (called response) and provides a challenge–response pair (CRP). Figure 2 shows the challenge–response protocol in our proposed model to generate an OPUF-based signature with the following CRP:

Challenge (*C*): the frequency-modulated continuous wave (FMCW) parameters on the Tx side represent the challenge. FMCW parameters are: sweep rate γ(Hz/s), which indicates the rate at which the frequency changed in FMCW; sweep range ΔF (ΔF = $\lambda_{stop}-\lambda_{start}$), which indicates the distance between the start and stop wavelength in FMCW; and the value of stop and start sweep wavelengths. Every single parameter has an important role in the spatial resolution and quality of the obtained RBS pattern; this fact is beyond the scope of this manuscript and is clarified in [35]. The required FMCW parameters to define a specific challenge to generate the signature are summarized in Table 1.

Response (*R*): the obtained RBS pattern represents the response. By changing FMCW parameters i.e., changing the challenge, the RBS pattern will change accordingly. Thereby, we can extract different RBS patterns from the same fiber if we stimulate the fiber with various challenges. Due to this fact, even if the adversary has access to the fiber, he cannot generate the intended signature without applying the correct challenge.

The procedure used to generate the signature is shown in Figure 3. Part of the obtained RBS pattern is extracted using a selecting window (SW). The measured data are converted to the binary domain as bit sequences, called the main sequence. Such operation is easily performed using a low-cost single-bit ADC. Subsequently, a sequence of random bits, called the key $K_{F1,F2}^{\left(1\right)}$, is generated and stored together with two parameters called flag points *F*1 and *F*2 that represent a specific random position in the key and main sequence, respectively. The main sequence and the key are then combined, as described in Figure 3c. The final sequence starts with the main sequence bits up to the *F*2 position followed by key bits up to the *F*1 position. The remaining bits are then combined and retain the same order: main sequence from the *F*2 position to the end followed by key bits from the *F*1 position to the end. It is notable that *F*1 and *F*2 are given as percentages of the corresponding sequence length.

The final sequence is converted into a quick-response (QR) code, i.e., a two-dimensional (2D) binary matrix, to denote the signature. This process has been performed through the MATLAB code by converting bits into the binary image. In the worst-case scenario, the adversary has physical access to the nominated fiber (or device) and possesses strong knowledge and high-performance devices that can predict the key. In this case, the adversary cannot find out the signature without knowing the key parameters. Consequently, adding a key with the mentioned parameters to the original data enhances the signature robustness against such adversary attacks. In Section 4.3, we will demonstrate which signature length, and consequently key length, can be defined due to the expected level of security from the signatures.

## 3. Signature Evaluation

We propose a new method to generate a signature associated to any optical fiber segment (i.e., optic links or devices’ fiber pigtails) to assist services including security, identification, authentication, and monitoring at the physical layer. However, the generated signatures must be unique and reproducible for every fiber. Thus, repeating measurements using the same challenge should result in the same signature. Moreover, it should be robust against physical and digital attacks. To address these significant features, some specific metrics must be considered.

In [31,32,33,34,35,36,37], we evaluated the obtained signatures’ uniqueness and reproducibility using Hamming distance. In this work, we further assess the signatures to complete the evaluation of the model’s performance. In addition to Hamming distance, we considered the following metrics, which have been used to assess the OPUF responses of the optical diffuser [38]:(A)Cross-Correlation Coefficient (XCOR-C):
(2)$$\mathrm{XCOR}-C\left(A,B\right)=\frac{\sum_{N}\sum_{M}\left(A_{mn}-\bar{A}\right)\left(B_{mn}-\bar{B}\right)}{\sqrt{\left[\sum_{N}\sum_{M}{\left(A_{mn}-\bar{A}\right)}^{2}\right]}\left[\sum_{N}\sum_{M}{\left(B_{mn}-B\right)}^{2}\right]}$$
where *A* and *B* are binary signatures with N×M dimension. The XCOR-C assesses the signatures based on the following procedures:shows how distinguishable the generated signatures from a single fiber (OPUF) are with different challenges;shows that noisy signature is distinguishable from an imposter version which is generated with the same OPUF but with the wrong challenge;assists in finding the best challenges to generate more reliable signatures.

The uniqueness assessment of the signatures is implemented through the XCOR-C. In this study, we have considered the Pearson cross-correlation coefficient, which can provide information regarding the structural similarity between the acquired signatures and accurately distinguish the noisy signature from the imposter. It is the most common way to assess the similarity between acquired PUF responses [38].

(B)Euclidean Distance (ED):

(3)$$\mathrm{ED}\left(\vec{x},\vec{y}\right)=||\vec{x}-\vec{y}||=\sqrt{\overset{N}{\underset{i=1}{\sum }}{\left(x_{i}-y_{i}\right)}^{2}}$$
where $\vec{x}$ and $\vec{y}$ are signature vectors (non-binary) with the length of N. The ED assesses the signatures based on the following:The ED between the RBS obtained through several measurements of the same OPUF and the same challenge translates into reproducibility;ED between the RBS obtained through several measurements of the same OPUF but different challenges indicates unpredictability;ED between the RBS obtained through several measurements of various OPUFs but with the same challenges indicates unclonability.
(C)Hamming Distance (HD):
(4)$$\mathrm{HD}\left(s1,s2\right)=\overset{M}{\underset{b=1}{\sum }}s1⊕s2$$
where s1 and s2 are the signatures vector (binary sequences) with the length of M. The HD shows the number of non-identical bits between two vectors, and if they get flipped, two signatures will be similar. The HD assesses the signatures based on the following:HD between the binary sequence achieved through several measurements of the same OPUF and the same challenge indicates reproducibility;HD between the binary sequence achieved through several measurements of the same OPUF but different challenges indicates unpredictability;HD between the binary sequence achieved through several measurements of various OPUFs but with the same challenges indicates unclonability;HD between each row of the binary image (QR code) gives uniqueness [37,38,39].

It is worth mentioning that the results of HD (ED) will commonly be demonstrated in histogram form [38]. In this manner, it is easy to find the HD (ED) threshold for accepting (genuine) or rejecting (imposter) the signature. Overlapping of the two histograms, which depict the HD (ED) between one OPUF response obtained through several measurements with the same challenge consisting of additional measurement noise (genuine signatures) and the HD between measurements of the same OPUF with different challenges (imposter), implies the presence of false positive (imposter signature accepted) or false negative (genuine signature rejected) measurements [31]. Zero overlapping guarantees that neither of the two challenges will result in an identical response, ensuring the signature’s unpredictability and unclonability.

(D) Statistical Test Suite (STS)

The aim of using STS is to run a test to investigate the probability of randomness of the bits in the signature sequences (S). Wherein, the hypothesis test result is returned as *h* = 1 or *h* = 0 with the probability of randomness *p*. If *h* = 0, the bits in S are in random order with the probability of *p*. Small values of *p* cast doubt on the validity of a null hypothesis. When *h* = 1, the test rejects the randomness of bits.

One of the most significant ways to explore a signature’s cybersecurity is by considering the random structure of the produced signatures (bits). In [32], we demonstrated the strength of the signature against a brute force attack using Hamming distance. Here, STS with *h* = 0 verifies the results in [32].

## 4. Security Validation

To validate the security robustness of the signatures generated using our proposed model, in terms of uniqueness, unclonability, and unpredictability, we considered various simulation scenarios for a 0.5 m fiber pigtail using C-OFDR. The measurement scenario was modified depending on which security feature of the signature was to be explored. In all scenarios, we considered a single polarization signal measure with C-OFDR.

### 4.1. Signature Evaluation through the ED and HD Metrics

To investigate the reproducibility and unpredictability of the signatures, which are defined in the following section, we considered an ADC with N = 4000 samples.

To validate robustness, we generated the signature by applying the following challenge: *Ci* (60 nm/s, 30 nm, 1530 nm, 1560 nm). The signature was stored as valid. Subsequently, we generated 100 signatures of the same fiber with the same challenge (*Ci*) adding random white Gaussian noise to emulate the signature robustness against measurement noise during different times of measurements. We implemented ED between the original signature and the noisy ones, as shown in Figure 4a.

To validate unpredictability, we generated the signature of the same fiber by applying 40 different challenges by changing the sweep rate (and sweep range in a few cases). One worst-case scenario of an adversary attack is when the adversary has access to the fiber and attempts to use different challenges to obtain the original signature. In essence, any new challenge generates a new response, even though the challenge is applied to the same PUF, and this fact neutralizes the adversarial attempts. To demonstrate that, we implemented ED between the original signature and the signatures obtained with different challenges. The results are depicted in Figure 4b. Figure 4 illustrates that there was no overlap between the robustness and unpredictability histograms, which diminishes the probability of false positives and false negatives. The histogram of robustness highlights a very low ED (around 0.9 on average), which means that the obtained responses are very similar notwithstanding the added noise. On the contrary, the histogram of unpredictability has higher values of ED (around 11 on average) by at least of one order of magnitude with respect to the previous case.

To validate unclonability, we generated 100 signatures from the 100 different fibers by applying the same challenge to all of them. Another worst-case scenario of adversary attacks is when the adversary knows the challenge but has no access to the original PUF and tries to apply the known challenge to different PUFs to find the signature. Because any PUF has a unique response for each specific challenge, the adversary trials will fail again. We implemented ED between the original signature and the other signatures obtained from other fibers with the same challenge as the original signature. Figure 5 shows no overlap between robustness and unclonability which reduces the probability of false positives and false negatives.

In all cases, the ED was implemented before the RBS conversion to the binary domain. To investigate the robustness, unpredictability, and unclonability of the signatures with the HD metric, we converted the abovementioned signatures to the binary domain, added the key, and generated the QR code. The normalized histogram of HD (expressed in percent) between the original signature and noisy signatures, and between the original signature and the signatures obtained with different challenges, is shown in Figure 6a,b, respectively.

The HD between the original signature and noisy signatures, and HD between the original signature and the other signatures obtained from other fibers with the same challenge as the original signature are shown in Figure 7a,b, respectively. The results of the HD metrics confirm the results obtained with the ED metric in the binary domain.

### 4.2. Signature Evaluation XCOR-C

Are noisy signatures adequately distinguishable from imposters?

To answer this question, we studied signatures through the XCOR-C metric. The original signature (QR code) was generated and stored in the database. The XCOR-C was, then, calculated between the original and noisy signatures. Meanwhile, different signatures (imposters) were generated by applying various challenges to the same OPUF (the original fiber pigtail). Subsequently, the XCOR-C between the generated signatures and the original signature was calculated. The obtained results indicate that different challenges on one OPUF make different signatures distinguishable from the original signal with the noise, as shown in Figure 8a.

Afterward, we dropped one noisy signature among the imposter signatures. Once more, we carried out the XCOR-C. The original noisy signal (genuine) was detected and appeared with the highest cross-correlation coefficient (peak) among the results. The results are shown in Figure 8b.

By regarding the obtained results, it is possible to define a threshold (TH) to accept or reject the signature as genuine. Consequently, the decision rule will be: if XCOR-C is below a certain threshold (here 0.5), the signature will be considered an imposter, but if it is above the TH, it will be deemed genuine.

### 4.3. Cyber Security

One of the most significant ways to explore a signature’s robustness against cyber attacks is to consider the random structure of the produced signatures (bits). Indeed, a sequence that consists of random bits is extremely strong against cyber attacks attempting to predict that sequence. To investigate the randomness of our proposed signatures, we generated 500 signatures (QR codes) with 2D and 60 × 60 bits, generated by 3500 samples and a key of 100 bits. After running the STS test among 500 signatures, only seven of them failed the test, i.e., 1.4% failed sequences. We increased the signature dimension by measuring the RBS with a higher number of samples (4000) and achieved a signature with 64 × 64 bits, consisting of a 96-bit key. The test showed only 0.6% failed sequences. Eventually, we increased the number of samples (5000) to increase the QR resolution to 71 × 71 bits, consisting of a 41-bit key. As a result, we obtained 0.0% failed sequences i.e., all signatures passed the test. The probability of randomness of signatures with different QR resolutions is depicted in Figure 9.

## 5. Discussion

Optical identification (OI) through an optical signature is a perfect security complementary tool to the networks and guarantees the security of systems, sub systems, and devices. Such an OI identification method may find uses within several applications. It can be used in applications related to classic network security [32], quantum network security [31], or network quality of transmission (QoT). Regarding a network device census, as highlighted in [40], it is common that network operators may not be fully aware of all the deployed fiber types in a network. This implies problems in the quality of transmission estimation, which increases estimation inaccuracy forcing the operator to assume even higher network margins than expected, with a consequent underestimation of the optical reach and an increase in the costs of regeneration [41]. Regarding network security, OI can be implemented for different network scenarios [32] to identify, authenticate, and monitor networks. As a simple example, we consider a point-to-point communication system that can be simply schematized through three sub-systems: transmitter, channel, and receiver. Each sub-system, as it was clarified in the previous sections, has its own signatures that are labeled as:

*Transmitter Signature*, *Channel Signature*, and *Receiver Signature*, respectively. In this scenario, when the channel is supposed to be a passive sub-system, three possible security approaches can be envisaged:the transmitter reads the *Channel Signature* and *Receiver Signature* to be sure that the information will pass to the specific channel and reach the specific receiver;the receiver reads the *Transmitter Signature* and *Channel Signature* so that the receiver knows the sender and the physical path;the transmitter reads the receiver, and the receiver acquires the *Transmitter Signature* so that both know with whom they are talking. Both may even check for *Channel Signatures* to check the path.

## 6. Conclusions

In this paper, the concept of a network digital signature has been introduced, in particular, it has been proposed to generate such a signature by resorting to using optical unclonable functions (OPUF) that are already intrinsically present in fibers and determined by the specific internal characteristics of each fiber. Specifically, the OPUF that has been individuated and then tested in this work is the Rayleigh backscattering signal (RBS). This kind of signal has been analyzed in order to understand if it is a good OPUF candidate that satisfies all the requirements that a PUF, in general, must fulfill to be adopted as a security mechanism in authentication scenarios. Such requirements are usually unclonability, robustness, unpredictability, and randomness; all of them have been thoroughly verified and evaluated on the basis of different metrics such as Hamming distance (HD), Euclidean distance (ED) and cross-correlation coefficient (XCOR-C). In all the different cases, the RBS has been demonstrated to significantly perform and has been confirmed to grant both robustness and distinctiveness.

The optical identification concept represents a novel approach to physical layer security which can be applied to any optical communication system and network. It provides supplementary quality and security to optical systems and network operations.

Future works will be dedicated to further investigating other typologies of metrics specific to real-valued signals, such as RBS, to better understand if superior performances can be achieved, particularly in terms of robustness. Following this line of investigation, diverse operative scenarios will be considered to comprehend the actual applicability of the proposed solution in more depth.

## Figures and Tables

**Figure 1 sensors-23-05269-f001:**
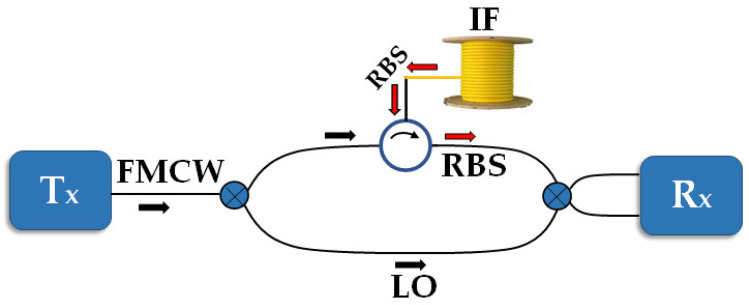
C-OFDR experimental setup. Tx: Transmitter (CW laser), FMCW: frequency modulated continuous wave, RBS: Rayleigh backscattering signal, LO: local oscillator, Rx: Coherent receiver.

**Figure 2 sensors-23-05269-f002:**
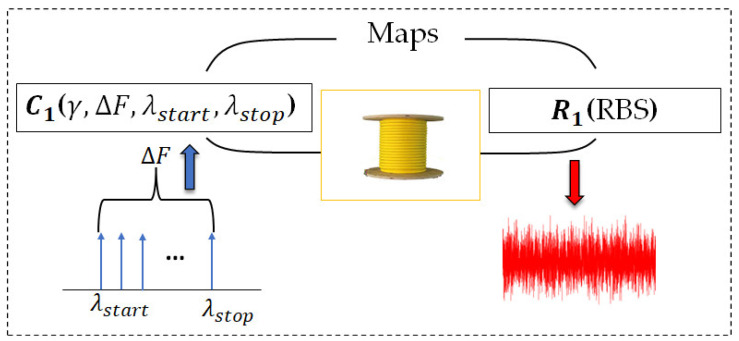
OPUF protocol to generate OPUF-based signature. Challenge 1 ($C_{1}$) ends with response 1 ($R_{1}$ ) which is a specific pattern of RBS of fiber (shown in red) associated with the specific challenge $C_{1}$.

**Figure 3 sensors-23-05269-f003:**
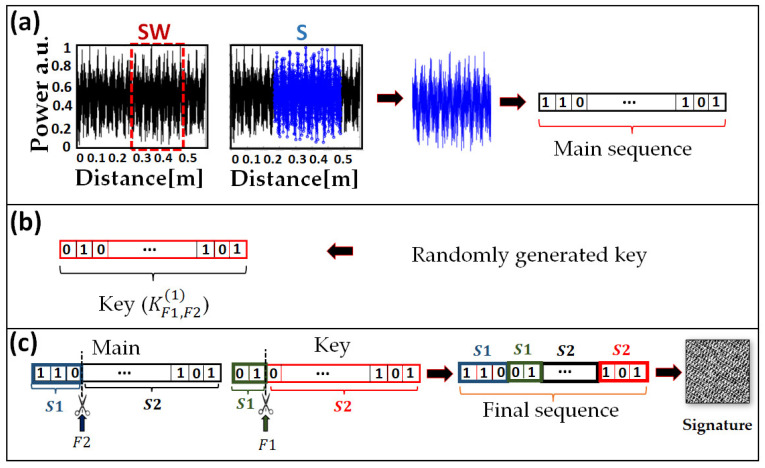
Post-processing for signature generation: (**a**) RBS pattern of 0.5 m fiber as a function of distance. SW: selecting window, S: selected data; (**b**) key sequence: $K_{F1,F2}^{\left(1\right)}$ the superscript is the challenge number, and the subscripts are the key’s parameters; (**c**) key and main sequence combination based on *F*1 and *F*2 flags to achieve the final sequence. The signature is obtained as a QR code from the final sequence.

**Figure 4 sensors-23-05269-f004:**
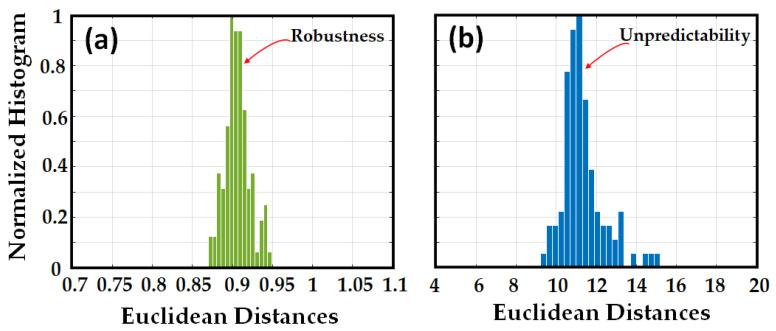
Normalized histogram of the Euclidean distances of signatures in the RBS stage, between (**a**) original signature (RBS) and 100 noisy signatures (RBSs), (**b**) original signature and 40 signatures (RBSs) generated from the original OPUF but with different challenges.

**Figure 5 sensors-23-05269-f005:**
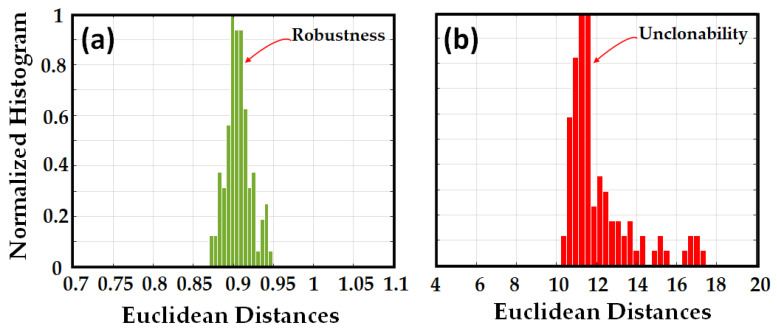
Normalized histogram of the Euclidean distances of signatures in the RBS stage between (**a**) original signature (RBS) and 100 noisy signatures (RBSs), (**b**) original signature (RBS) and 100 signatures (RBSs) generated from different OPUFs (100) but with the same challenge.

**Figure 6 sensors-23-05269-f006:**
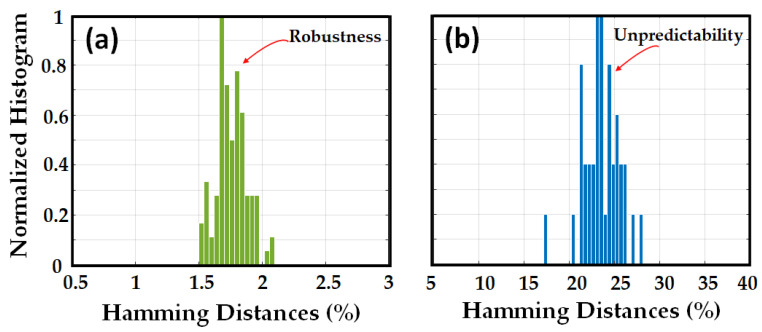
Normalized histogram of the Hamming distances for signatures in the binary domain, between (**a**) original signature and 100 noisy signatures, (**b**) original signature and 40 signatures generated from the original OPUF but with different challenges.

**Figure 7 sensors-23-05269-f007:**
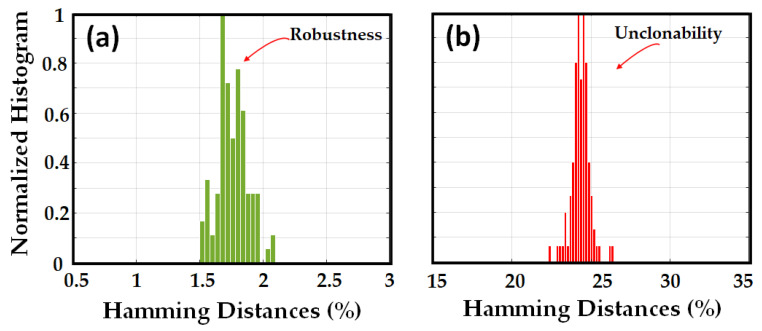
Normalized histogram of the Hamming distances for signatures in the binary domain, between (**a**) original signature and 100 noisy signatures, (**b**) original signature and 100 signatures generated from different OPUF (100) but with the same challenge.

**Figure 8 sensors-23-05269-f008:**
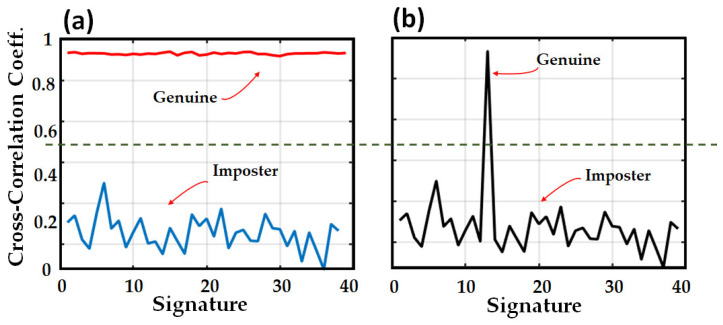
Cross-correlation coefficient between original signature, (**a**) and noisy signatures (red) and the imposter signatures (blue). (**b**) A random collection of signatures, among which the genuine signature is detected (peak). Defined threshold: green dashed line.

**Figure 9 sensors-23-05269-f009:**
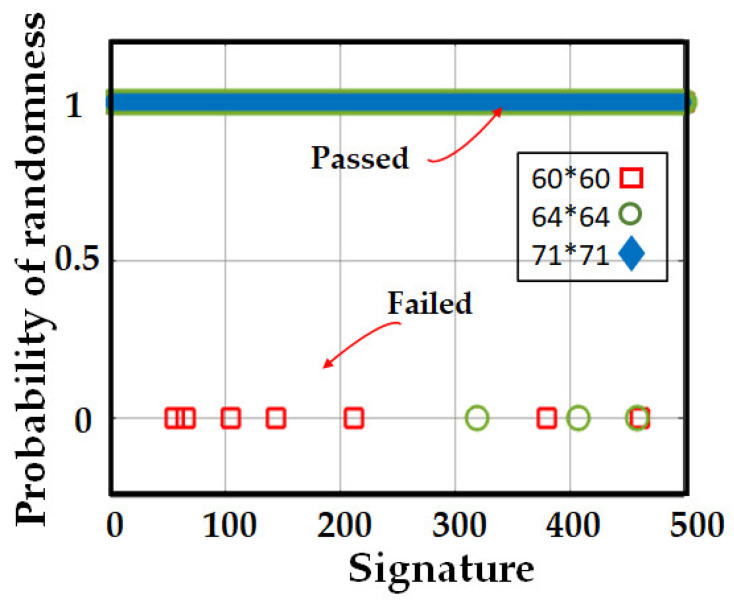
Randomness test over 1500 signatures.

**Table 1 sensors-23-05269-t001:** Required FMCW parameters to define a challenge to generate the signature.

Symbol	Description	Unit
γ	Sweep rate	nm/s (or Hz/s)
ΔF	Sweep range	nm (or Hz)
$\lambda_{\mathrm{start}}$(or $f_{\mathrm{start}}$)	Start sweeping wavelength (or frequency)	nm (or Hz)
$\lambda_{\mathrm{stop}}$(or $f_{\mathrm{stop}}$)	Stop sweeping wavelength (or frequency)	nm (or Hz)

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
