# Peer review of "Optical Systems Identification through Rayleigh Backscattering"

_sensors, 2023, doi:10.3390/s23115269_

Round 1

Reviewer 1 Report

Dear authors,

Structure and content of the paper are at a very good level. All the results are thoroughly described.

Nevertheless, I have some questions and comments: 

1. In formula (1) argument of cosine function must be dimensionless and Intensity must be proportional the squre of the filed amplitude E. As far as I can see, these conditions are not met now.

2. What is the typical length of signature sequences and what is the typicl length of the key (page 5)?

3. What is the procedure of convertation of final sequence to QR-code? (string 179). Could you please insert a reference or described the procedure briefly?

5. Ref.19 is not full (it is probably a typo) 

6. What is the CMOS in string 75?

Reviewer 2 Report

The authors present a rather interesting article, this area of research is really relevant. The introduce a technique to generate and read out the digital signature of the networks, channels, and optical devices that possess the fiber optic pigtails to enhance physical layer security (PLS). They claim that attributing a signature to the networks or devices eases the identification and authentication of networks and systems thus reducing their vulnerability against physical and digital attacks. Despite the fact that the topic of identifying a fiber optic line by an OFDR is already proposed, this manuscript contains important practical results. However, I have a number of questions that need to be answered before publishing this article:

- Firstly, the experimental setup is not described in details. Any experiment presented in a scientific paper must be repeatable. This would be great if you provide the setup detailed data and present it as a table. Namely:

A) FMCW - please give the key parameters (model or parameters from the datasheet), line width (coherence length), sweeping range and rate.

B) Since a high-coherence version of the setup is used, such a setup will not operate correctly due to the lack of a clock- or auxiliary-interferometer that provides compensation of the laser sweeping nonlinearity. The effect of its use is clearly presented in the recent work [10.3103/S1062873822700599] and in these two earlier papers [10.1364/OE.17.005845], [10.1364/OE.20.028319]. Please clarify this.

C) Unfortunately, the text of the work does not mention anywhere the influence of the polarization state of the radiation on the results obtained. However, it is known that random fluctuations of the polarization state occurring in isotropic media (like typical SMF) can significantly distort the OFDR-trace. Thus, the same optical link under slightly different conditions can produce different back reflections, even if no attacks occur. In the presented setup, there are no polarizers or polarization beam splitters dividing the signal for two separate photodiodes, which usually simplify the solution of this problem. The setups like this are presented in [10.3390/s20226407], [10.1117/12.2229057]. I ask the authors to clarify this point too.

D) How do you control the absolute wavelength of the laser? A gas cell can be used for the wavelength accuracy improvement to sub pm [10.3390/s18103480], but I can not see anything to solve this problem in the setup. Or these measures are excess for this study? This is a very important question too.

- the authors use the Pearson cross-correlation coefficient, which gives more complex and resource-intensive calculations than obtaining an ordinary cross-correlation function in a discrete form. Could the authors explain in the text why this way was chosen?

- I would ask the authors to highlight the practical results obtained more clearly.

- Some sections of the article end with figures, I would authors to place them higher, immediately after the first mention in the text.

Round 2

Reviewer 2 Report

I thank the authors for their accurate work and answers. I think the paper could be published now.